# A signal peptide peptidase is required for ER-symbiosome proximal association and protein secretion

Jian Yang [1], Niu Zhai [2], Yuhui Chen [3], Luying Wang [1], Rujin Chen [3] & Huairong Pan [1] ✉

During legume-rhizobia symbiosis, differentiation of the symbiosome (engulfed intracellular rhizobia) is necessary for successful nitrogen fixation. To control symbiosome differentiation, host cell subcellular components, e.g., ER (endoplasmic reticulum), must adapt robustly to ensure large-scale host protein secretion to the new organelle. However, the key components controlling the adaption of ER in nodule cells remain elusive. We report that Medicago *BID1*, a nodule-specific signal peptide peptidase (SPP), is central to ER structural dynamics and host protein secretion. In *bid1*, symbiosome differentiation is blocked. BID1 localizes specifically to the ER membrane and expresses exclusively in nodule cells with symbiosomes. In the wild type ER forms proximal association structures with symbiosomes, but not in *bid1*. Consequently, in *bid1* excessive ER stress responses are induced and ER-to-symbiosome protein secretion is impaired. In summary, a nodule-specific SPP is necessary for ER-symbiosome proximal association, host protein secretion, and symbiosome differentiation.

The nitrogen-fixing symbiosis (NFS) between legumes and rhizobia ensures plants an efficient way to get fixed nitrogen. During NFS, plants form a specific organ, the root nodule, to host their microbial partners. Inside nodules, rhizobia fully enter host cells, are encapsulated by a plant-derived lipid membrane, and form a subcellular organelle-like structure, the symbiosome (rhizobia within symbiosome is termed bacteroid)[1].

Once fully settled inside host cells, symbiosomes surrender the control of their fate to hosts, to a great degree. In Inverted Repeat-Lacking Clade (featured by the loss of a 25-kilobase inverted repeat in the chloroplast genome) species such as the model legume *Medicago truncatula*, symbiosomes undergo terminal differentiation, which is characterized by drastic changes in genome ploidy, morphology, transcriptional, metabolic patterns and lost abilities of propagation[2]. As a prerequisite for successful NFS (free-living rhizobia and undifferentiated symbiosome cannot fix nitrogen), symbiosome differentiation is initiated and regulated by host secreted proteins[2,3]. Large numbers of host-secreted proteins have been identified. In Medicago, host cells secrete NCR (Nodule-specific Cysteine Rich) peptides, nodule-specific GRP (Glycine Rich Peptides), and many others[4-8], to promote symbiosome differentiation. High-throughput trafficking of host proteins is controlled by a nodule-specific protein secretory pathway[3]. Through alternative cleavage and polyadenylation, host cells generate a nodule-specific isoform of SYP132 (SYNTAXIN 132), which marks the symbiosome membrane, the interface between host cell and bacteroid, to guide host protein trafficking[9,10]. Several key regulators of host-to-symbiosome vesicle trafficking have also been reported[11,12]. Considering the importance of symbiosome differentiation to this symbiosis, research into nodule cell-specific protein secretion is critical to our understanding of the NFS.

ER (endoplasmic reticulum) is a sub-cellular organelle central to intracellular protein production and secretion. Its structure is highly dynamic, moreover, the ER forms direct membrane-to-membrane

[1]College of Biology, Hunan University, Changsha, China. [2]Zhengzhou Tobacco Research Institute of CNTC, Zhengzhou, China. [3]College of Life Sciences, Lanzhou University, Lanzhou, China. ✉e-mail: hrpan@hnu.edu.cn

structures for interaction with other organelles, including mitochondria, Golgi, and the plasma membrane, to transport lipids and other molecules[13,14]. During NFS, the ER plays a key role in accommodating and communicating with symbiosomes, with the aforementioned host protein secretion as the most prominent example. While symbiosome-destined host proteins are diverse in their sequences and structures, most of them have a N-terminal signal peptide (SP) in their nascent sequences, which is cleaved by the DNF1 nodule-specific Signal Peptidase Complex (SPC) at the ER membrane, to facilitate proper folding and secretion[6,15]. More than controlling protein production and secretion, early studies suggest that ER tubes are spatially close to membranes of differentiated symbiosomes[16], and there are reports of ER structural adjustments in rhizobial infection and symbiosome development[17–19]. The mechanism beneath robust ER adaption to assert the delicate control of host protein secretion and host-symbiosome communication remains elusive. A more specific question is related to SPs, upon SP excision, it remains unknown how nodule cells cope with SP fragments on ER membrane, to ensure proper ER function and protein secretion.

Here we report that in nodule cells, a nodule-specific signal peptide peptidase (SPP) at the ER membrane is necessary for proper ER structural reconfiguration, successful host protein secretion, and NFS.

## Results

### Symbiosome differentiation is blocked in *bid1*

To find key genes regulating symbiosome differentiation, we screened for mutants with a "*bacteroids with impaired differentiation*" phenotype (*bid* mutants). *Medicago truncatula* line NF-FN6798 (named *bid1* here), a fast neutron bombardment mutant from the Noble Research Institute, the United States, had white, round and small nodules (Fig. 1a, Supplementary Fig. 1a). In *bid1*, nodule cells were smaller, symbiosomes were undifferentiated and less orderly arrayed (Fig. 1b, Supplementary Fig. 1b, c). Confocal microscopy and TEM assays confirmed that in *bid1* cells, rhizobia within symbiosomes failed to differentiate (Fig. 1c, d, Supplementary Fig. 1d), reminiscent of *dnf1* defects[15]. Furthermore, in the analysis of isolated symbiosomes, indeed *bid1* symbiosomes were smaller and less elongated (Fig. 1e), their length in *bid1* was much lower compared to WT (Fig. 1g). Consistent with defects in differentiation, in *bid1* the expression levels of *NCR001* and *NCR169*, two representatives of differentiation-promoting NCR peptides, were much lower (Fig. 1h, Supplementary Fig. 1e). Moreover, expression levels of several key rhizobial genes were altered in *bid1* nodules. *exoY* encodes an enzyme required in early steps of exopolysaccharide synthesis[20]; *ctrA* is a key transcription factor controlling bacterial cell cycle[21]; and *bacA* encodes a membrane protein protecting rhizobia from antimicrobial molecules, e.g., NCR peptides[22]. The expression levels of *exoY*, *exoB*, *ctrA* and *bacA* were much higher compared with WT (Fig. 1i, j, Supplementary Fig. 1f, g), indicating altered states of exopolysaccharides synthesis and impaired differentiation of bacteroids in *bid1* cells[23–25]. Consequently, rhizobial *nifH::GUS* reporter was not activated in *bid1* nodules (Fig. 1f). These results show that in *bid1*, symbiosome differentiation is blocked at an early stage upon symbiosome formation.

### *BID1* encodes a nodule-specific SPP

To identify the *BID1* gene, we performed a bulked segregation assay using the F2 population of *bid1* crossed to WT A20. *bid1* mutation was mapped to the upper arm of Chromosome 1, close to marker 001e11 (Fig. 2a). Subsequently We performed whole genome sequencing of *bid1*. In *bid1*, a 8857 bp fragment from Chromosome 2, 44,572,921 to 44,581,777 bp, is translocated onto Chromosome 1, next to the adenine at 877,946 bp (Fig. 2a). The translocation is within the 5' UTR region of MtrunA17_Chr1g0147151 (Medtr1g008280 in Mt4.0), at 102 bp upstream of the start codon (Fig. 2a). The expression of target gene was significantly reduced (Supplementary

Fig. 1h), proving translocation-caused transcriptional inactivation. Expressing the genomic sequence of the target gene with a N-terminal GFP tag in *bid1* could complement the "fix-" phenotype (Fig. 2b), confirming *bid1* phenotype is caused by the defect in MtrunA17_Chr1g0147151 gene.

*BID1* is annotated to encode a SPP, an aspartic endopeptidase with intra-membrane protein cleavage activity[26,27]. As a critical component for ER protein secretion, SPP digests remnant SPs inside the ER membrane upon SPC-mediated release of SPs from nascent polypeptides[28,29]. BID1 protein contains 8 transmembrane domains, with two conserved aspartate residues in the middle of the 5th and 6th transmembrane, respectively (Fig. 2c)[29]. The C-terminus of BID1 contains a conserved KKXX motif, which is critical for ER-membrane localization of many ER resident proteins (Fig. 2c)[30]. By searching the Alphafold2 protein structure database[31], we found the predicted BID1 structure was identical to that of human SPP protein (Supplementary Fig. 1i), further confirming BID1 functions as a SPP.

Key genes regulating symbiosome differentiation are usually nodule-specifically expressed[32]. A β-glucuronidase (GUS) reporter assay was performed to investigate the expression pattern of *BID1*. *pBID1::GUS* activity was detectable only in nodules, in every nodule zone except meristem, indicating *BID1* is a nodule-specific gene (Fig. 2d), which was also confirmed by analyzing public RNA-Seq and Microarray data (Supplementary Fig. 2a, b). Moreover, *BID1* is expressed in close correlation with *DNF1*, a representative of nodule-specific SPC (Supplementary Fig. 2a-c)[15], similar *cis* elements were identified in promoter regions of *BID1* and *DNF1* (Supplementary Fig. 2d, 2e). The expression pattern of *BID1* is consistent with its fundamental roles in symbiosome differentiation, also transcriptionally it may be activated together with *DNF1*.

BID1 is a multi-pass membrane protein. We expressed *GFP-BID1* fusion sequence driven by *35 S* promoter together with *mCherry-HDEL*, the widely used ER marker[33], in *Nicotiana benthamiana* leaves. GFP-BID1 had an ER web-like distribution pattern and co-localized with mCherry-HDEL (Fig. 2e, g), showing BID1 probably localizes at ER membrane. To confirm BID1 localization in nodule cells, we transformed *bid1* with a construct expressing *GFP-BID1* fusion sequence under *BID1* promoter and Arabidopsis *UBQ10*-driven *mCherry-HDEL* reporter together. This construct could also complement *bid1* phenotype (Supplementary Fig. 2f). Under confocal microscopy, GFP-BID1 co-localized well with mCherry-HDEL (Fig. 2f, h), proving BID1 is an ER membrane protein in nodule cells. When GFP was fused to BID1 at the C-terminus, BID1-GFP could not complement *bid1* (Fig. 2b). The KKXX motif is required strictly at the C-terminus for successful ER membrane localization[30], fusing GFP to the C-terminus would have blocked proper BID1 targeting.

In *GFP-BID1* nodules, GFP signals could only be detected in large cells (Supplementary Fig. 2g). As symbiosome-containing cells tend to be larger, *BID1* may only exist in cells infected by rhizobia. When *GFP-BID1* plants were inoculated with Rm1021 mCherry strain, indeed GFP-BID1 was specifically restricted to cells containing symbiosomes (Supplementary Fig. 2h), proving *BID1* expresses exclusively in infected cells.

### *BID1* evolves from a housekeeping homolog

Key to the ER protein secretory apparatus, SPP is highly conserved among yeast, mammalians and plants[29]. As *BID1* only expressed in nodules (Fig. 2d), we searched for its housekeeping homologs. In Medicago, *BID1* has one homolog with a highly similar sequence (Supplementary Fig. 3a), which we named *BID1-Like* (*BID1L*). The gene structure of *BID1L* is also highly similar to *BID1* (Supplementary Fig. 3b). Unlike *BID1*, *BID1L* expression is similar across Medicago tissues (Supplementary Fig. 3c). *BID1* probably evolved as a nodule-specific SPP basing on *BID1L*. To confirm this notion, we constructed a

phylogenetic tree of *BID1* and *BID1L* paralogs from moncots, dicots, including multiple legumes[1]. While BID1L paralogs could be found in every species (some have more than one copy), BID1 paralogs were specific to IRLC-clade species (Supplementary Fig. 4). This is similar to nodule-specific DNF1[15], indicating in IRLC-clade legumes, genes function in secretory pathway evolved nodule-specific duplicates to promote terminal symbiosome differentiation.

## ER stress responses are induced in *bid1*

As a SPP, BID1 functions to cleave SP fragments on ER membrane. Missing *BID1* will probably cause accumulation of SP fragments, leading to excessive ER stresses and disturbance of ER functions[34–36]. To determine ER stress responses in *bid1*, we checked the expression levels of ER stress marker genes, which tend to have multiple orthologues in Medicago (Supplementary Fig. 5a). Majority of ER stress marker genes were significantly activated in *bid1* nodules (Fig. 3a-e, Supplementary Fig. 5b-g). Orthologues of *bZIP28* and *bZIP60*, two basic leucine zipper (bZIP) transcription factors[37–39], were highly induced (Fig. 3a-c). *BiP3* (*bZIP induced protein 3*) orthologues[40,41], the ER-resident HSP70 cognates activated by *bZIP28* and *bZIP60*, were also highly expressed in *bid1* (Figs. 3d, e). Surprisingly the expression levels of *BAG7c* (homolog of *Bcl-2-associated athanogene 7*)[42], and *ERO1* (*Endoplasmic Reticulum Oxidoreductions 1*)[43], two genes necessary for proper maintenance of ER stress responses, were downregulated (Supplementary Fig. 5d, e). We reason that in *bid1* cells, ER stress responses are activated to some extent, in a unique way. How *BID1* mutation leads to specific ER stress responses requires further investigation.

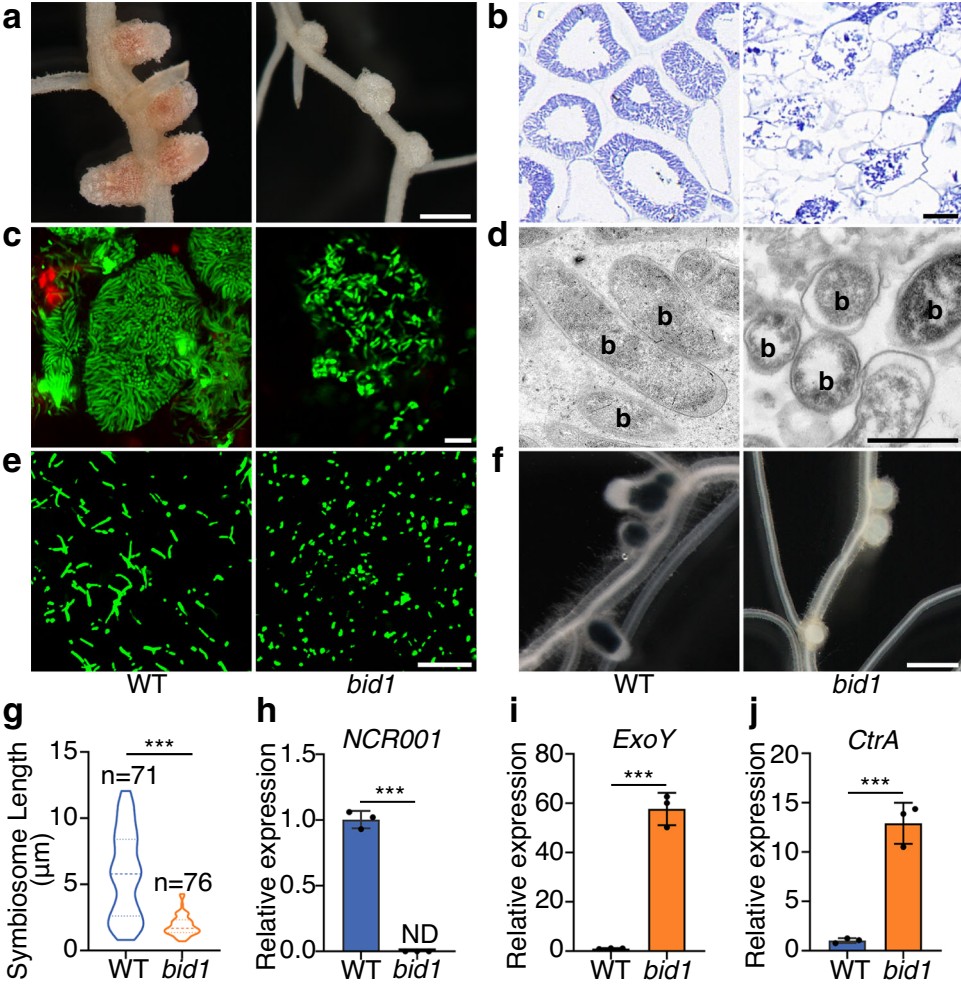

**Fig. 1 | Symbiosome differentiation is impaired in nodule cells of *bid1*. a** *bid1* mutant only had small and white nodules when inoculated with *S. meliloti* ABS7 *hemA::LacZ*. Representative pictures of nodules were taken at 21 dpi (days post inoculation). Bar=1 mm. **b** Symbiosomes were not differentiated in *bid1* nodule cells from **a** in nodule sectioning assay. Note that infected *bid1* nodule cells were also smaller than WT. Nodules from **a** were sectioned into 5 µm slides and stained with toluidine blue. Bar=20 µm. **c** Symbiosomes in *bid1* nodule cells were undifferentiated. 21 dpi nodules inoculated with *S. meliloti* RM021 *pHC60-GFP* were stained with PI (Propidium iodide, red) and analyzed under confocal microscopy. Bar=10 µm. **d** Symbiosomes were undifferentiated in *bid1* nodule cells in TEM assay. 21 dpi nodules inoculated with ABS7 *hemA::LacZ* were used for analysis. Letter "b" indicated bacteroids. Bar=2 µm. **e** Representative pictures showing differences between isolated symbiosomes from WT and *bid1*. Symbiosomes were isolated from nodules in **c**. Bar=10 µm. **f** *bid1* symbiosomes were much shorter in length in the quantification assay of symbiosomes from **e**. Piano plot to show the minimum, 25th percentile, median, 75th percentile and maximum symbiosome lengths. Numbers of symbiosome analyzed were indicated respectively. "***", *P* < 0.0001 in one-way ANOVA assay. **g** *nifH::GUS* reporter was not activated in *bid1* nodule cells. GUS staining was performed on WT and *bid1* nodules 21 days after inoculation with *S. meliloti* RM1021 *nifH::GUS* at 21 dpi. Bar=750 µm. **h**, Expression of *NCR001* was totally blocked in *bid1* nodule cells. qRT-PCR assay was performed using samples from nodules 21 dpi with ABS7 *hemA::LacZ*. ND, not detected. **i** and **j**, *Rhizobium ExoY* and *CtrA* genes were highly expressed in *bid1* nodule cells compared with WT. qRT-PCR assays were done using the same samples as in **h**. For **h**, **i** and **j**, "**" and "***" meant *P* < 0.01 and *P* < 0.001 in Student's *t*-test, respectively. Data were represented as means ± SEM of three independent amplifications. Experiments were repeated at least 3 times with similar results.

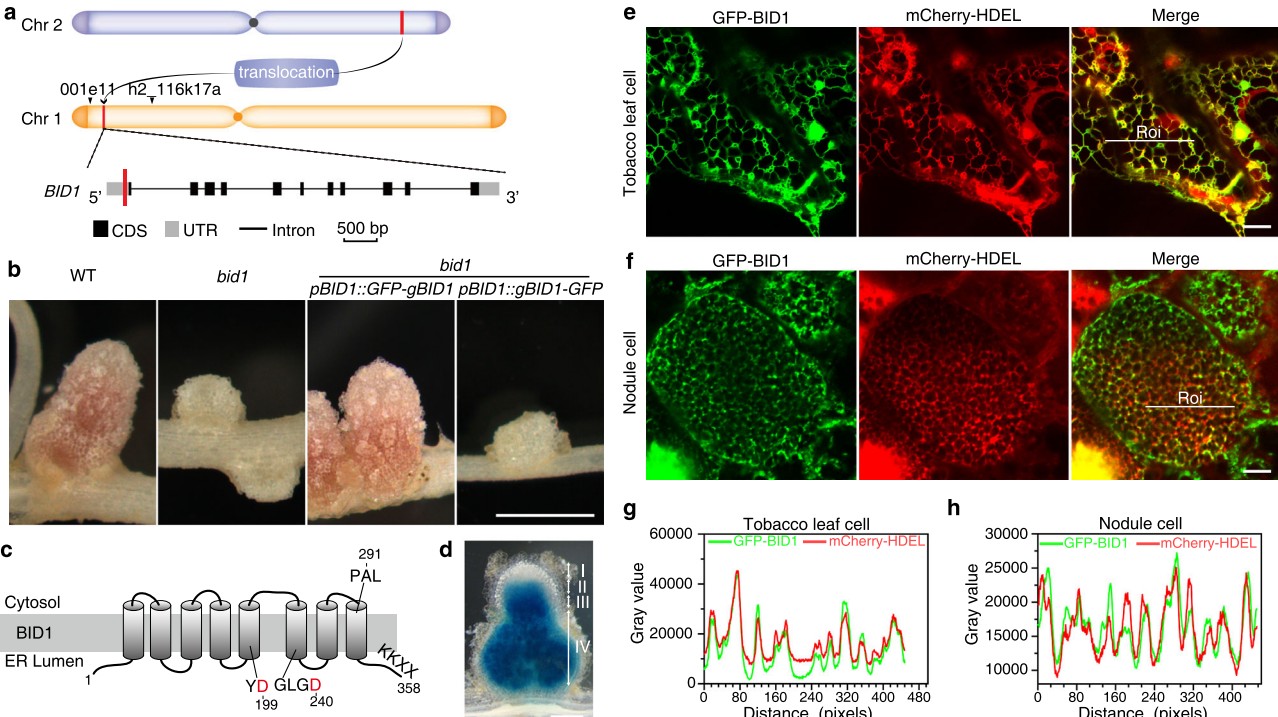

**Fig. 2 | *BID1* encodes a nodule-specific, ER membrane-localized SPP. a** A diagram showing *BID1* gene structure and the position of the target mutation in rough mapping. *BID1* was mapped to the upper arm of Chromosome 1, in the interval between markers 001e_01 and h2_116k17a. A 8,857-bp fragment from Chromosome 2, 44,572,929-44,581,777 bp, is inserted into the 5′ UTR of *MtrunA17_Chr1g0147151*, 102 bp ahead of the start codon. **b** Expressing *GFP*-fused candidate gene genomic sequence driven by its own promoter could complement the *bid1* phenotype. Please note that GFP tag need be inserted at N-terminus of BID1. Bar=2 mm. **c** BID1 is a multi-transmembrane domain-containing SPP protein. The transmembrane domains were shown, and two conserved aspartate residues critical for peptidase activity were labeled red. PAL, a conserved motif important for peptidase activity; KKXX, the C-terminal ER membrane localization signal. Numbers indicated positions of the amino acids. **d** *BID1* is expressed exclusively in nodules in promoter-GUS reporter assay. Zone I, nodule meristem, Zone II, infection zone, the zone where rhizobia infect nodule cells, Zone III, differentiation zone, the zone where symbiosomes undergo differentiation, Zone IV, fixation zone, the zone symbiosomes fix nitrogen. Bar=500 μm. **e** BID1 localized specifically at ER membrane when expressed in the leaves of *Nicotiana benthamiana*. *N. benthamiana* leaves co-transformed with *35 S::GFP-BID1* and *mCherry-HDEL* ER reporter were analyzed under confocal microscopy at 2 days post-transformation. Bar=10 μm. **f**, BID1 localized to the ER membrane in nodule cells. A construct expressing *pBID1::GFP-BID1* together with *mCherry-HDEL* reporter was transformed into *bid1* mutants. Bar=10 μm. **g** and **h** Fluorescent intensity measurement of GFP-BID1 and mCherry-HDEL in indicated regions of tobacco leaf and nodule cells respectively. Fluorescent peaks of GFP-BID1 overlapped with mCherry-HDEL signals in both cell types. Fluorescent intensity was measured as grey value of pixels using ImageJ software. Roi, region of interest. Experiments were repeated more than 3 times with similar results.

## ER-symbiosome proximal associations are impaired in *bid1* cells

ER is a large, membranous organelle functioning in protein and lipid production and secretion, its structure is highly dynamic and it can have distinct shapes[13]. To investigate disturbances to ER functions by undigested SP fragments and activated ER stress responses, we checked ER structures in WT and *bid1* nodule cells. When stained by ER-Tracker Red, a chemical that stains ER membrane specifically, in infected WT cells, ER was highly expanded during cell development, and ER signals were in proximity with symbiosomes, a strong indication of ER structural remodelling during symbiosome differentiation (Supplementary Fig. 6). In *bid1*, the ER was arranged irregularly throughout the nodules (Supplementary Fig. 6), indicating failed ER structural reconfiguration.

We further determined ER structure using an mCherry-HDEL reporter. Overall ER structure varied significantly between WT and *bid1* cells (Supplementary Fig. 7a). In zone-by-zone comparisons, only in the infection zone were ER structural patterns similar between WT and *bid1* (Supplementary Fig. 7b-h). In WT differentiation and fixation zone cells, mCherry-HDEL labeled ER formed web-like structures, which expanded extensively and were in close proximity to GFP-labeled symbiosomes, while in *bid1*, mCherry-HDEL signals were dot-like, and the ER failed to form regularly arrayed webs around symbiosomes (Fig. 3f-h, Supplementary Fig. 7b-h). Neither WT nor *bid1* uninfected cells form an extensive ER web in ER-tracker Red staining or ER fluorescent reporter assays, indicating cell type-specificity of ER structures (Supplementary Fig. 8a, b).

A TEM assay was performed to investigate the presumed ER-symbiosome association more thoroughly. Corroborating with confocal microscopy, in WT cells, there were a large number of ER sheets following symbiosomes closely (Fig. 3i). In *bid1*, relatively straight lines of ER were lost, ER structures were disordered and failed to associate closely with symbiosomes (Fig. 3i), and ER width and area size were larger (Fig. 3k, l). Moreover we utilized serial AutoCUTS-SEM (Scanning Electronic Microscopy) and 3D topological reconstruction to build ER and symbiosome structural models. Notably in WT nodule cells, symbiosomes were completely encircled by dense ER lines (Fig. 3j, Supplementary Movie 1, 3), with an average distance between the ER and symbiosome of several dozen nanometers (Fig. 3n), well within the range of organelle membrane-membrane contact[13,14]. In *bid1* cells, the ER was not structured as well-arrayed sheets and failed to surround symbiosomes (Fig. 3j, Supplementary Movie 2, 4), the ER volume was much lower (Fig. 3m), and average ER-symbiosome membrane distances were much larger (Fig. 3n), indicating damaged ER structural construction and ER-symbiosome proximal association. Our results suggest that the ER of nitrogen-fixing nodule cells is closely associated with symbiosomes, and that this requires BID1.

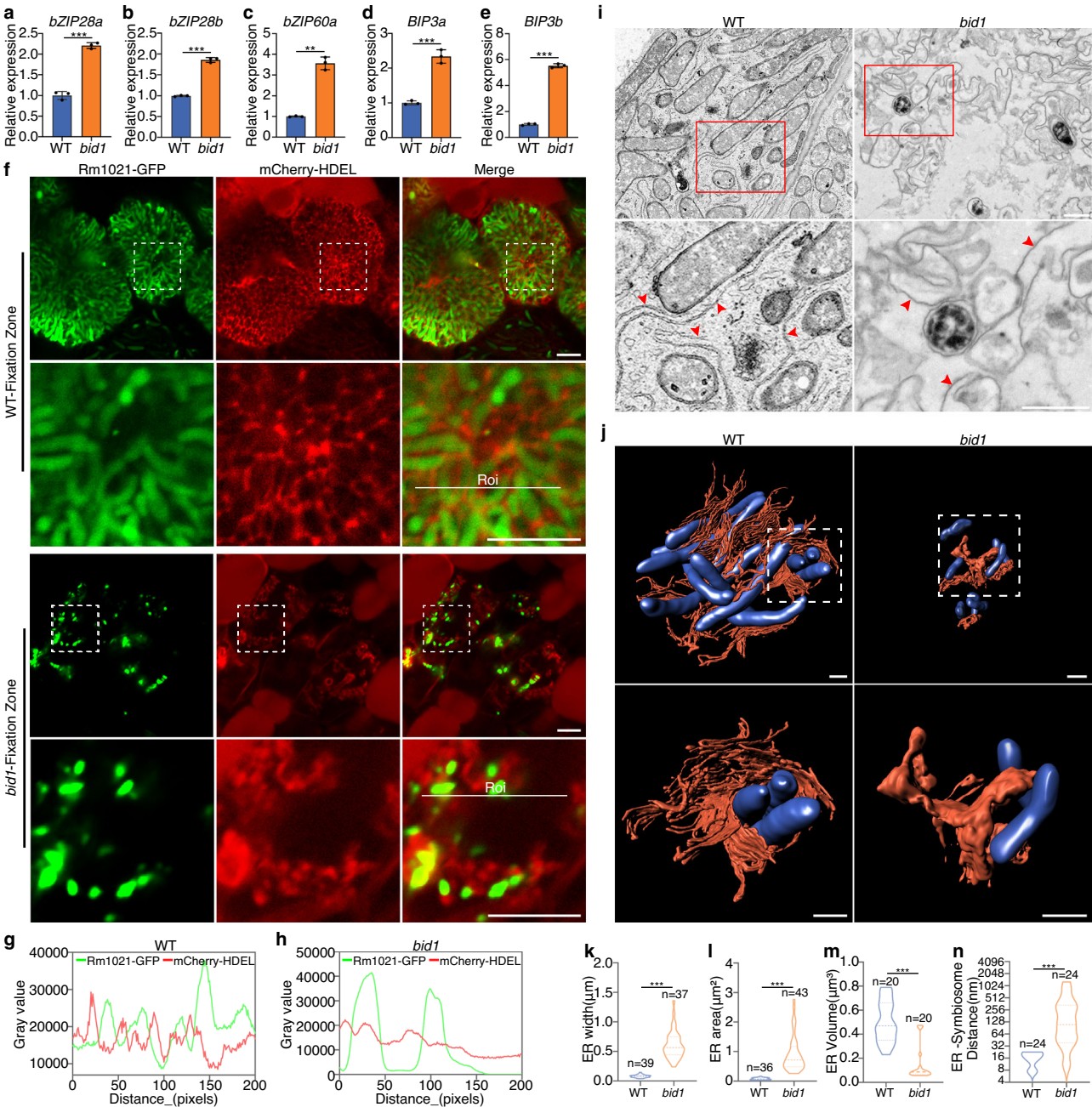

**Fig. 3 | BID1 is required for proper ER-symbiosome proximal association. a-e** In qRT-PCR assay a group of ER stress-related marker genes were induced in 21 dpi *bid1* nodules inoculated with ABS7 *hemA::LacZ*. Error bars represent standard deviation from three biological replicates. "**" and "***", *P* < 0.01 and *P* < 0.001 in Student's *t*-test respectively. Data were represented as means ± SEM of three independent amplifications. **f** Confocal microscopy analysis of ER structures in WT and *bid1* nodule cells. 14 dpi Rm1021 *pHC60-GFP* inoculated mCherry-HDEL-expressing WT and *bid1* nodules were analyzed. For both panels, bar=10 μm. Roi, region of interest. **g** and **h** Fluorescent intensity of symbiosome GFP and ER mCherry signals of indicated regions in WT and *bid1* respectively. Fluorescent intensity (grey value of pixels) was measured by ImageJ. **i** TEM assay results of ER and symbiosome structures in 14 dpi ABS7 *hemA::LacZ*-inoculated WT and *bid1* nodule cells. For both panels, bar=1 μm. **j** Reconstructed 3D structures of symbiosomes and ER in WT and *bid1* cells varied significantly. Overview of several ER and symbiosomes and zoomed-in structures of individual symbiosome and ER were shown. Symbiosomes were labelled in blue, ER in orange red. Structures were obtained through 3D reconstruction of AutoCUTS-SEM tomography data using Imaris software. Scal bar, 1 μm. **k** and **l** Piano charts showing differences in ER width and area between WT and *bid1*, respectively. ER width and area in **i** were measured using ImageJ. "***", *P* < 0.001 in One-way ANOVA assay. **m** Measurement of ER volumes in WT and *bid1* by Imaris. "*", *P* < 0.05 in one-way ANOVA assay. Numbers of ER analyzed were indicated. **n** Average distance between ER membrane and symbiosome was much larger in *bid1*. ER-symbiosome distances were measured using Imaris. y-axis, logarithmic scores of ER-symbiosome distance to the base 2. "***", *P* < 0.001 in one-way ANOVA assay. For **k, l** and **n**, numbers of ER analyzed were indicated. For **m**, 3D ER structures were evenly divided into 20 segments, volumes of each segment were determined. Experiments were repeated 3 times with similar results.

## BID1 is required for host protein secretion

The dissociation of ER and symbiosomes in *bid1* suggests that host protein secretion may be impaired. To test this, we investigated ER-to-symbiosome secretion of several known host proteins. NCR001, a representative substrate of the DNF1 nodule-specific SPC[6], NCR166, another NCR peptide expressed early during symbiosome differentiation (Supplementary Fig. 9a), and CAML1 (*Calmodulin-Like 1*), a member of the nodule specific calmodulin-like protein subfamily[5],

were expressed in WT and *bid1*, respectively, under the *BID1* promoter. *BID1* promoter could activate the GUS reporter in *bid1* mutant (Supplementary Fig. 9b), indeed expression levels of *NCR001-GFP*, *NCR166-GFP* and *CAML1-GFP* transcripts were even higher in *bid1* than WT (Supplementary Fig. 9c). Confocal microscopy revealed that while in WT GFP-tagged NCR001, NCR166 and CAML1 localized to the symbiosome, likely the peri-bacteroid space, in *bid1* these proteins could not be secreted (Fig. 4a, d, g). Similar results were obtained by analyzing individual symbiosomes (Fig. 4b, e, h), further demonstrating blocked host protein secretion in *bid1*. Confocal microscopy results were confirmed by Western blot analyses of isolated symbiosomes (Fig. 4c, f), although we failed to detect CAML1-GFP. These results show that in *bid1* mutant, host protein secretion is blocked, showing degrading SP fragments on the ER membrane is necessary for proper ER-to-symbiosome protein trafficking.

## Discussion

In this work we report BID1, a nodule cell-specific SPP functioning at the ER membrane, is critical for symbiosome differentiation in nodule cells. Beyond illustrating SPP-mediated nodule-specific protein secretion (Fig. 4), the discovery of *BID1* has broader impacts for our understanding of ER stress responses and ER-symbiosome proximal association during NFS (Fig. 5).

The Arabidopsis genome has one SPP, which is lethal if knocked out[34]. This is probably also be true for *BID1L*, the housekeeping SPP in Medicago, and SPP orthologues from other plant species. Thus *bid1* holds unique and irreplaceable advantages in studying molecular links between mutations in SPP and ER stress responses, as in the mutant ER stress responses are activated in a unique way (Fig. 3a-e, Supplementary Fig. 5b-g).

ER structural adaption during symbiosome differentiation has been hinted long time ago[16–19], yet the structural relationships between the ER and symbiosome are not clear. In maturing cells with symbiosomes, the ER forms an extensive web-like structure spreading throughout the cytoplasm (Fig. 3f-i). Using AutoCUTs-SEM and 3D modelling, we show that the ER structure is closely associated with differentiating symbiosomes (Fig. 3j), with the distance between ER and symbiosome well within the range of organelle contact (Fig. 3n). In *bid1* nodule cells, with the SPP missing, ER structure is significantly altered, the well-arrayed ER web, and in most cases, the ER-symbiosome proximal associations, are missing (Fig. 3i-n), highlighting the impacts of BID1 on ER-symbiosome structural association. As exemplified by BID1 (Fig. 3f-n), there might be a complete set of nodule-specific genes, which acquired their functions from their housekeeping homologs, that function in ER structural construction.

The possible functions of ER-symbiosome proximal association are intriguing. The ER can form a close membrane-membrane interaction with every organelle and plasma membrane in various cell types, their direct interaction can facilitate trafficking of lipids and other molecules[13,14]. In infected cells of the fixation zone, mature symbiosomes occupy a large proportion of the cell volume (Figs. 1c, d, 3f), ER-symbiosome proximal association may facilitate the transfer of lipids and other metabolites. For instance, since the bacteria are enveloped by a lipid membrane derived from plants[11], certain lipids may be transferred to symbiosome through ER-symbiosome proximal association, as is the case of phosphotidylethanolamine transportation between ER and mitochondria in mammalian cells with the nutrients depleted[44]. Combining this work and earlier studies about ER adaption during NFS[16,17,19], it is highly possible the ER and symbiosome form direct organelle contact structures in nodule cells (Fig. 3f-n). To fully prove direct organelle contact between the ER and symbiosome, a tethering complex and its cargos should be identified. Nevertheless existence of the contact can be demonstrated, at least structurally, by findings of BID1-mediated ER-symbiosome proximal associations (Fig. 3f-n, Supplementary Fig. 7).

Beyond signal peptide fragments, multiple substrates of human SPP have been reported[45,46]. It should not escape our attention that some cleaved products from BID1-mediated degradation of SP fragments may have novel functions in nodule cells, and BID1 may have a broader spectrum of substrates beyond SP, similar to scenarios in human cells[47,48]. Further investigations into BID1, particularly through the identification of its direct substrates by biochemical assays, will significantly contribute to fully revealing the molecular communications between nodule host cells and their microbial partner in the symbiosomes.

## Methods

### Plant growth, rhizobia inoculation and hairy root transformation

*Medicago truncatula* A17 and A20 plants were used in this study. Plants were grown in growth rooms at 22 °C, with 16 hours light (150 μE m$^{-2}$ sec$^{-1}$) and 8 hours dark. *S. meliloti* strains ABS7 *hemA::LacZ*, Rm1021 *nifH::GUS*, Rm1021 *mCherry* and Rm1021 *pHC60-GFP* were used for plant inoculation in this study. In brief, rhizobia collected from fresh overnight liquid culture were suspended in half basic nodulation medium (BNM), to the concentration of OD$_{600}$ = 0.05. 5 mL of liquid BNM culture were used for inoculation per plant in green zeolite. *Agrobacterium rhizogenes* strain Arqua1 was used for hairy root transformation of Medicago plants, the experiments were performed following previously described procedures[49]. To select transformed plant tissues, seedlings infected by agrobacteria strains were grown in Fahraeus medium containing 15 μg/mL kanamycin or 25 μg/mL hygromycin (antibiotics were chosen according to antibiotic resistance genes of the specific constructs) for 10 days.

### Map-based cloning, whole genome sequencing and complementation assays

To map *BID1* gene, *bid1* mutant was crossed with WT A20 plants. F1 plants were inoculated with ABS7 *hemA::LacZ* to prove *bid1* is a recessive mutation. In F2 population, seedlings were also inoculated with ABS7 *hemA::LacZ*. At 21 dpi, leaf samples from fix- and fix+ plants were collected respectively and subjected to bulked segregation analysis, using a group of genetic markers to determine the rough location of *BID1*. Upon mapped *BID1* to a unique locus on Chromosome 1, DNA samples were sent to BGI Shenzhen for whole genome sequencing using BGI sequencing platform. Whole genome sequencing data were analyzed using IGV software (Integrative Genome Viewer, https://software.broadinstitute.org/software/igv/).

Two constructs were used to complement *bid1* mutant with the target gene, *pBID1::GFP-gBID1-pKGW-RR* and *pBID1::GFP-gBID1-UBQ10::mCherry-HDEL-pKGW*. The constructs were transformed into *bid1* mutants through hairy root transformation, transformed plants were inoculated with ABS7 *hemA::LacZ* and phenotypes were determined at 21 dpi.

### Molecular cloning

In this study KOD Plus DNA polymerase (TOYOBO, Japan) was used to amplify target DNA fragments. The ligations were performed through In-fusion cloning using ClonExpress UltraOneStep Kit (Vazyme, China). For *pBID1-pMDC163* construct, a 2.5 Kb fragment containing *BID1* promoter and 5' UTR was ligated into *pMDC163* directly. To make the *pBID1::GFP-gBID1-pKGW* construct, *GFP* coding sequence was ligated into *pUC18* vector, then the DNA fragment containing *BID1* promoter and 5' UTR and *BID1* genomic sequence were ligated into *GFP-pUC18* respectively. Then the *pBID1::GFP-gBID1* sequence was amplified through PCR and ligated into *pKGW*. To make the *pBID1::GFP-gBID1-UBQ10::mCherry-HDEL-pKGW* construct, *mCherry-HDEL* was amplified and ligated with *UBQ10* promoter, then the complete *UBQ10::mCherry-HDEL* sequence was amplified and ligated into *pBID1::GFP-gBID1-pKGW*. Meanwhile the *UBQ10::mCherry-HDEL* fragment was ligated into *pKGW*

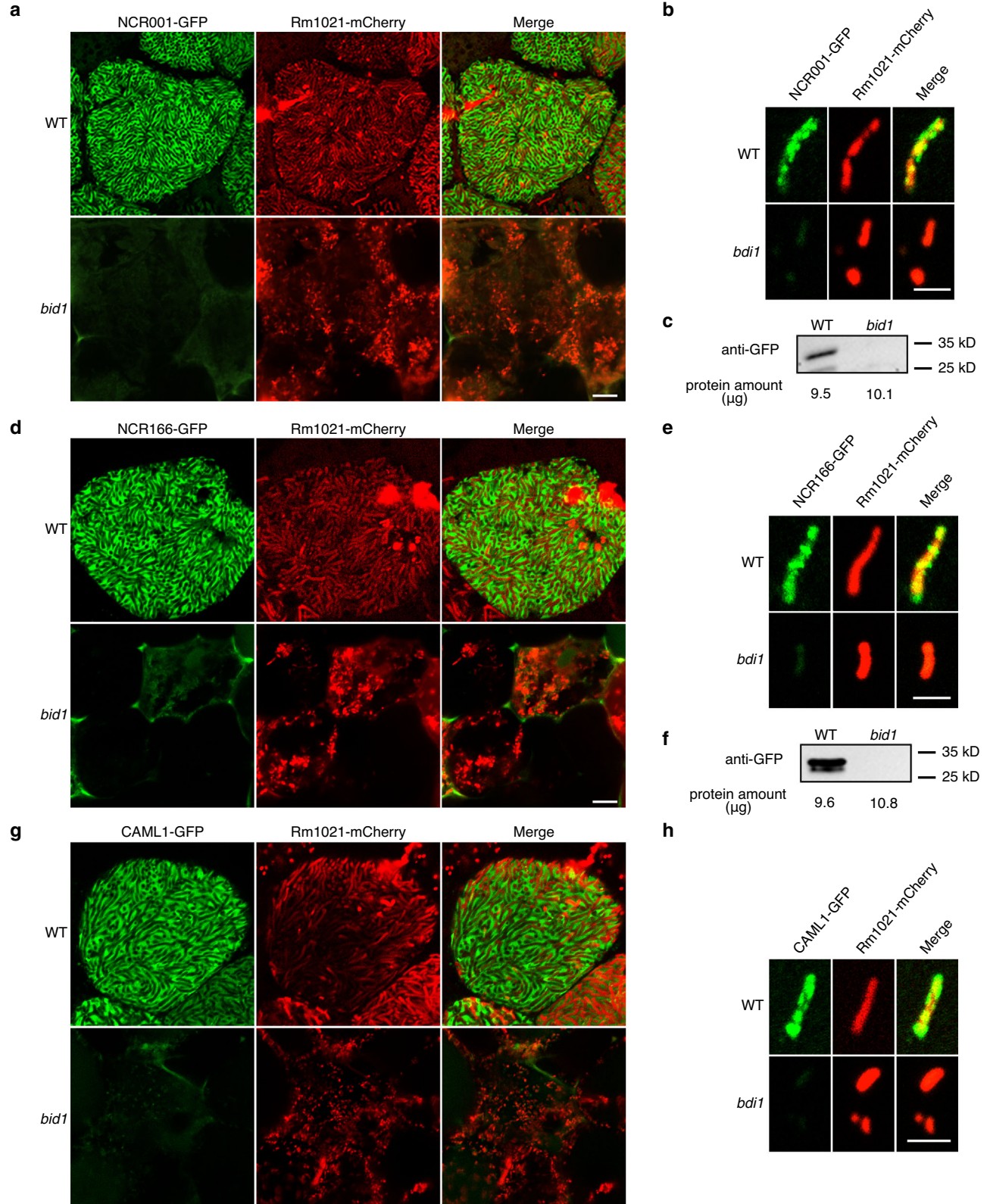

**Fig. 4 | BID1 is required for the secretion pathway in nodule cells. a, d** and **g** Secretion of NCR001, NCR166 and CAML1 were blocked in *bid1* nodule cells. *GFP*-fused *NCR001*, *NCR166* and *CAML1* were expressed under the *BID1* promoter in WT and *bid1*, respectively. 14-day-old nodules inoculated with *S. meliloti* Rm1021 mCherry were analyzed by confocal microscopy. Bar=10 μm. **b, e** and **h** GFP-fused NCR001, NCR166 and CAML1 could not be detected in *bid1* symbiosome. Symbiosomes were isolated from nodules in **a, c**, and **e** respectively, and analyzed by confocal microscopy. Bar=5 μm. **c** and **f**, In Western blot assay of isolated symbiosome, secretion of NCR001 and NCR166 were blocked in *bid1* nodule cells. *GFP*-fused *NCR001*, *NCR166* were expressed under *BID1* promoter in WT and *bid1*. Symbiosomes were isolated from 14-day-old nodules inoculated with ABS7 *hemA::LacZ*. Protein samples were subjected to Western blot. Protein amounts loaded in each sample were labeled. Experiments were repeated more than 3 times with similar results.

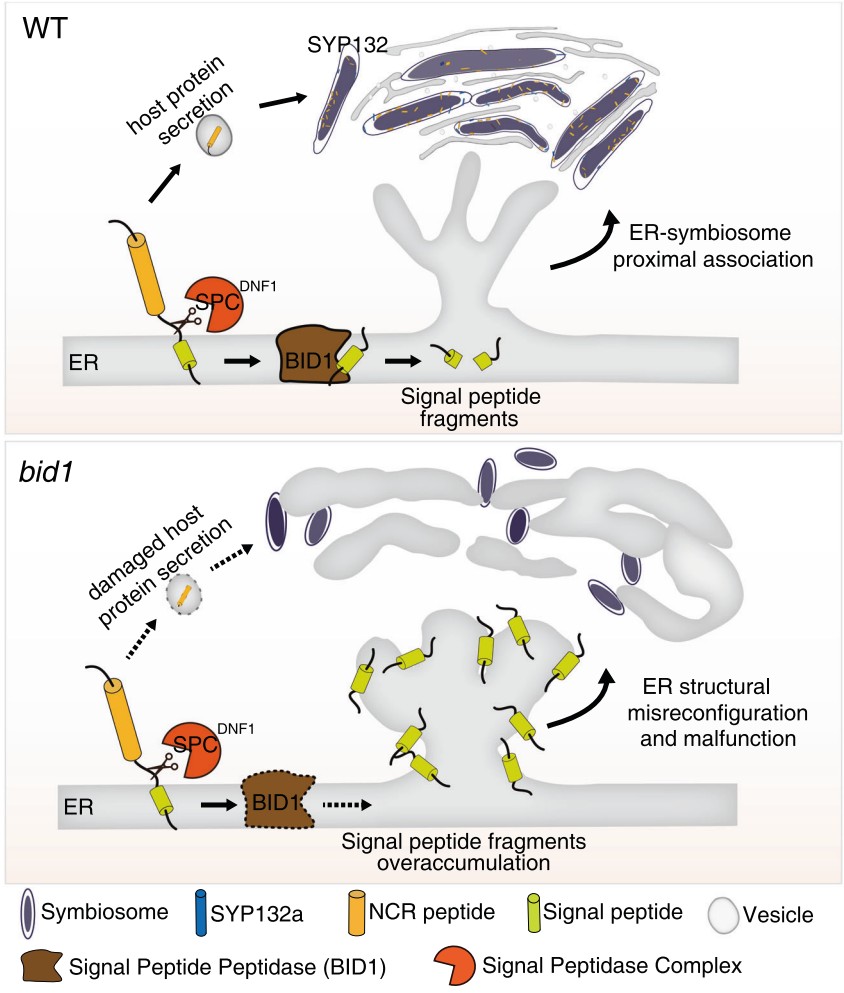

**Fig. 5 | Diagram to summarize the function of BID1 in nodule cell ER reconfiguration and protein secretion.** Nodule cells secrete a large number of proteins, e.g., NCR peptides, to symbiosomes to promote their differentiation. SP sequences could be found at the N terminus of these host proteins. The DNF1 nodule-specific SPC cleaves SP fragments from nascent sequences of host proteins to ensure proper folding and SYP132-mediated target protein secretion. On the ER membrane, remnant SP fragments will be cleaved by BID1, the nodule-specific SPP. In WT cells containing symbiosomes, ER forms an extensive web-like structure, closely surrounding differentiated symbiosomes. In *bid1* mutant, with the nodule-specific SPP missed, ER structural reconfiguration and ER-symbiosomes proximal association are damaged, resulting in blockage of host protein secretion. In summary the nodule-specific SPP on ER membrane is critical for ER structural reconfiguration, ER-symbiosome proximal association, and host protein secretion.

directly to generate the *mCherry-HDEL* reporter construct. For *pBID1::gBID1-GFP* construct, a DNA fragment covering *BID1* promoter to the stop codon was amplified and ligated into *pMDC107*, then the *pBID1::gBID1-GFP* fragment was amplified and ligated into *pKGW*. To express *GFP*-fused *NCRO01*, *NCR166* and *CAML1* in nodule cells, genomic sequences of the genes were amplified and ligated into *pBID1-pKGW* (a DNA fragment containing *BID1* promoter and 5′ UTR sequence was ligated into *pKGW* in advance) respectively. All the constructs used in this study were listed in Supplemental Table 1, primers used for molecular cloning were listed in Supplemental Table 2.

### Nodule sectioning, toluidine blue staining, X-gal and X-Gluc staining

To section and stain the nodules, freshly picked nodules were vacuumed for 1 hour and fixed with 0.05 M nitrate phosphate buffer, pH 7.2, which contains 4% (w/v) paraformaldehyde and 5% (v/v) glutaraldehyde, for 12 hours at 4 °C. Fixed nodules were then dehydrated in the ethanol/dimethylbenzene solution series, and were embedded in paraffin subsequently. Then the samples were sliced into 5 μm thick sections on the Biosystems RM2245 microtome (Leica, Germany).

Postsectioning, nodule slides were stained in 0.05% toluidine blue solution for 20 mins, and washed thoroughly in ddH$_2$O. Stained nodule slides were checked and pictured using 3D Histech Pannoramic MIDI Slide Scanner (3DHISTECH, Hungary) following manufacturer's instructions. X-Gal and X-Gluc staining were performed as previously described[50], briefly fresh nodules were cut in half and stained with 0.8 mg/mL X-gal in Z′ buffer, or GUS staining buffer containing 10 mM/L X-Gluc respectively, for 12 hours at 37 °C. Samples were then washed with 100% ethanol until at least the roots were clear. The stained samples were analyzed and pictured, on the TL500 microsystem (Leica, Germany) equipped with a DMC6200 digital camera.

### Confocal microscopy analysis

For confocal microscopy, nodules were hand-sectioned in half, the products were sectioned again for the observations. To determine symbiosome phenotypes through SYTO9 and PI staining, the sectioned nodules were stained with 10 mM/L SYTO9 and 5 mM/L PI for 5 mins. The excitation/emission wavelengths for SYTO9 and PI were set at 483 nm/503 nm and 493 nm/636 nm, respectively. For GFP and mCherry fluorescence, the excitation/emission wavelengths were 488 nm/507 nm and 587 nm/610 nm, respectively. For ER-Tracker Red

staining (Beyotime, China), nodule samples were stained for 15 mins in the staining solution, the excitation and emission wavelengths were 587 nm and 615 nm, respectively. Samples were observed under a Ti-E + A1 MP Confocal Laser Scanning Microscope (Nikon, Japan).

## Quantitative Real-time PCR

Quantitative Real-time (qRT) PCR was performed as previously described[51], in brief total nodule RNA was extracted using TRIzol (Life Technology, the United States). Extracted RNA were treated with Turbo DNA-free kit (Invitrogen, the United States) to eliminate genomic DNA contamination. iScript cDNA synthesis kit (Bio-Rad, the United States) were used to synthesize the cDNA. All of the above procedures were done following manufacturers' instructions. cDNA were diluted to a concentration of 0.5 μg/μL prior to PCR. qRT-PCR experiments were performed at a volume of 20 μL/reaction with 1 μg cDNA template. Experiments were performed using PerfectStart® Green qPCR SuperMix (Transgen, #AQ601-01) running on a Mastercycler EP Realplex system (Eppendorf, Germany). To rule out any non-specific amplifications, melting curves of PCR products were determined. Results were represented as means of threshold cycle values of three replicates. Relative expression of Medicago genes was normalized to *PDF2* (MtrunA17_Chr6g0484701), for rhizobial genes, 16 S rRNA was used as the internal control. Primers used for qRT-PCR were listed in Supplemental Table 2.

## Symbiosome protein extraction and proteomics assay

Extraction of symbiosome proteins were performed as previously described[52]. In brief, freshly collected nodules were put into the pre-cooled (4 °C) extract solution I containing 0.5 M sucrose, 10 mM DTT, 50 mM Tris-HCl (pH 7.4), and 1% (v/v) protease inhibitor cocktail (P8340, Sigma-Aldrich), and were grounded thoroughly. Then samples were filtered through 2 layers of Microcloth (Calbiochem, the United States). Part of the filtered samples were centrifuged at 14,000 rpm for 20 mins and the supernatant was designated as total nodule proteins. For cytosol proteins, samples postfiltering were centrifuged at 10,000 g for 1 min, the supernatant was collected and labeled cytosol proteins. Then the pellets above centrifuge were re-suspended in a new solution with 1.5 M sucrose, 10 mM DTT, 50 mM Tris-HCl (pH 7.4) and 1% protease inhibitor cocktail (v/v), and centrifuged at 10,000 g for 90 seconds. The new pellets were re-suspended in extract solution I and labeled symbiosome proteins. The concentration of protein samples were determined by Pierce™ BCA Protein Assay Kit according to manufacturers' instructions. Equal amount of protein samples were boiled in 4× Laemmli buffer and were separated on SDS-PAGE gels. Proteins were then transferred to nitrocellulose membranes (Adventec) for immunoblotting analysis. The membrane was incubated with anti-GFP primary antibody (1:2000, GFP-Tag(7G9) mAb, Abmart) and secondary antibody (1:2500, Goat Anti-Mouse IgG HRP, Abmart), respectively and signals were visualized by enhanced chemiluminescence (Thermo Scientific) on an integrated chemiluminescent gel imaging system (OI600 Touch, Bio-OI).

## AutoCUTS-SEM sample preparation, serial sectioning, microscopy and 3D reconstruction

14 dpi WT and *bid1* nodules inoculated with ABS *hemA::LacZ* were fixed 0.1 M Phosphate Buffer (pH 7.4) with 2.5% glutaraldehyde (v/v) and 2% paraformaldehydewith (v/v), and washed twice in Phosphate Buffer and ddH$_2$O respectively. Fixed nodules were immersed in 1% OsO$_4$ (w/v) and 1.5% K$_3$Fe(CN)$_6$ (w/v) aqueous solution at 4°C for 1 h, washed, incubated in filtered 1% thiocarbohydrazide aqueous solution (Sigma-Aldrich) at 25°C for 30 min, 1% unbuffered OsO$_4$ aqueous solution at 4°C for 1 h, and 1% UA aqueous solution at 25°C for 2 h. Then nodules were dehydrated with alcohol solution series (30, 50, 70, 80, 90, 100%, 100%), and pure acetone (twice), 10 min each at 4°C, infiltrated in graded mixtures (8:1, 5:1, 3:1, 1:1, 1:3, 1:5) of acetone and SPI-PON812

resin (21 mL SPI-PON812, 13 mL DDSA, 11 mL NMA, 1.5% BDMA), finally pure resin. Next nodules were embedded in pure resin (21 mL SPI-PON812, 13 mL DDSA, 10 mL NMA, 1.5% BDMA) and polymerized for 12 h at 45 °C, followed by 48 h at 60 °C. For 3D ultrastructure study, sections (449 for WT, 354 for *bid1*, respectively) with thickness at 50 nm were prepared using ultramicrotome (UC7, Leica, Germany) with the AutoCUTS device (Zhenjiang Lehua Technology, China). Serial sections were automatically acquired by Helios Nanolab 600i dual-beam SEM (Thermo Fisher, the United States) with AutoSEE as the imaging software. Image parameters were set as follows, accelerating voltage 2 kV, beam current 0.69 nA, CBS detector, pixel size 6.7 nm for A17, 5.4 nm for *bid1*, dwell time 3 microseconds. Alignment of serial sections was done by the registration function of MiRA-Align, all datasets were analyzed by Imaris software (Version 9.2.1).

## Statistics and reproducibility

All quantitative data were presented as Mean ± S.D. The *P* values were calculated by Student's t-test (unpaired and two-tailed) assays using GraphPad Prism 8 and the Excel software. A *P* value lower than 0.05 was considered statistically significant. "*" indicated $P < 0.05$, "**", $P < 0.01$, "***", $P < 0.001$, "****", $P < 0.0001$, respectively. All experiments were replicated at least three times, representative data from the independent experiments were shown. No specific randomization or blinding protocols were used.

## Reporting summary

Further information on research design is available in the Nature Portfolio Reporting Summary linked to this article.

## Data availability

Source Data generated in this study are provided in the Source Data file, and have also been deposited in the Figshare database under accession code https://doi.org/10.6084/m9.figshare.23077898. The gene expression data of plant and rhizobial genes mentioned in this study were obtained from Medicago Symbimics Database (https://iant.toulouse.inra.fr/symbimics/) and Medicago Gene Expression Atlas (https://medicago.toulouse.inrae.fr/MtExpress). Additional information is available from the corresponding author upon request. Source data are provided with this paper.

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

## Acknowledgements

We thank Prof. Lijing Liu from Shandong University, China for the mCherry-HDEL marker construct, Prof. Zhaosheng Kong from Institute of Microbiology, Chinese Academy of Sciences for *S. meliloti* Rm2011 DsRed strain. We thank Dr. Xixia Li, Center for Biological Imaging (CBI), Institute of Biophysics, Chinese Academy of Sciences, for her help of AutoCUT-SEM assays. We thank Prof. Dong Wang, University of Massachusetts Amherst, for insightful discussions of the manuscript. This work is funded by National Natural Science Foundation of China grants 31870221, 32070271 and 32161133006, and Changsha Science and Technology Program grant kq2004026 to H.P.

## Author contributions

H.P. and J.Y. designed the research; J.Y. and N.Z. performed most of the experiments; H.P., J.Y. and N.Z. analyzed the experimental results; Y.C. and R.C. built the Medicago fast neutron bombardment mutant library and identified the initial FN6798 mutant; L.W. helped with molecular cloning and hairy root transformation experiments; H.P. wrote the manuscript with inputs from other authors.

## Competing interests

The authors declare no competing interests.
