## [Peer Review File · Nature Communications]

REVIEWER COMMENTS

Reviewer #1 (Remarks to the Author):

This manuscript explores the role of BID 1, a putative peptide peptidase, suggested to be involved in cleaving signal peptides in the ER. A genome rearrangement mutant, bid1, was isolated and nicely used to illustrate the consequences of impaired BID1 function in ER dynamics, protein secretion and nodule function.

The strength of this research is that it represents a novel approach to understand ER-symbiosome association and the role of BID1 in protein secretion required for symbiosome maturation. A major weakness is that the central claim of BID1 being a peptide peptidase is only supported by indirect evidence like sequence homology and AF2 modelling. Biochemical results supporting the peptidase activity of BID1 would make the story substantially stronger.

Expression levels of ExoY, ExoB, CtrA and BacA were higher in bid1 mutants (Fig 1) and this is suggested to indicate a free living state of the bacteroids. Not sure what is the evidence for this higher expression is related to the free-living state. This statement needs to be substantiated or supported by references if this has been shown before.

Reviewer #2 (Remarks to the Author):

This manuscript describes a signal peptide peptidase required for nodule function. The phenotype is very strong, with blocked symbiosomes differentiation and N-fixation. The cell biology is of high quality, and the most important conclusions are strongly supported (ie. that in bid1 nodules the ER is dissociated from the symbiosomes). The nodulation phenotype presumably results to a defect in ER function resulting either from the accumulation of signal peptides in ER. The gene is single copy in Arabidopsis its loss is lethal. Hence, BID1 represents a unique opportunity to study signal peptide peptidases in plants, an issue that is discussed by the authors.

I have only a few minor points and a list of suggested edits which mostly concern grammar. The authors conclude since only legumes with indeterminate nodules have BID1 that genes function in secretory pathway evolved nodule-specific duplicates frequently. However, they show that BID1 is limited to a specific legume clade, strongly suggesting that BID1 evolved once, in the common ancestor of the IRLC clade. The conclusion is therefore incorrect, at least based on the data presented. Also, mimosas, a non-IRLC- legume, form indeterminate nodules. Is BID1 present in mimosas?

minor comments:

The thickness of optical section used for confocal images should be noted, since when considering associations between the ER and bacteroids, thick sections could be misleading.

In Fig 5, is the signal peptide usually ER membrane associated? If so this should be shown in the diagram.

The authors say that the GFP tag needed be inserted at N-terminus of BID1. You must provide details; did you try C-term and it failed?

In extended data Fig 1, indicate what ND stands for

In Extended data Fig9c, were these genes expressed from pBID1? Please make this more clear. Extended In Extended Data Figure 4. The species name is missing for lcu_2RBY

Abstract line 27: implications of what?

29: to the ER membrane

30: the ER

31: but not in bid1

58: of the NF symbiosis

62 membrane-to-membrane structures for interaction with other

62 and the plasma membrane

63 the ER

64 delete aforementioned or use 'the aforementioned'

67 at the ER

69 studies suggest that

70 adjustments during rhizobial

73 to assert the delicate control of

74 to SPs, upon SP excision, it remains unclear how nodule cells cope with SP fragments on the ER membrane.

87 membrane is necessary (delete comma)

87 Confocal microscopy and TEM assays confirmed that in bid1 cells, rhizobia within symbiosomes failed to differentiate

154 In Medicago, BID1 has one homolog with a highly similar sequence.

154 Provide the gene model for BIDL

160 delete specifically

161 This is similar to nodule-specific DNF1

184 in bid1 nodule cells

187 In bid1, the ER

187 Suggested rewording: In bid1, the ER was arranged irregularly throughout the nodule (Extended Data Fig. 6).

190 an mCherry

190 structure instead of arrangement

192 only in the infection

194 and were in close proximity to GFP-labeled symbiosomes,

195 and the ER failed to form regularly arrayed webs around the symbiosomes.

197 form an extensive

203 structures were disordered and failed to associate closely with symbiosomes

207 with an average distance between the ER and symbiosomes of several dozen nanometers

208 refs 13,14 belong is the corresponding section in the discussion

209 failed to surround [encircle is 2D, surround is 3D]

212-13 Suggested wording: Our results suggest that the ER of N-fixing nodule cells is closely associated with symbiosomes, and that this requires BID1.

216 The dissociation of ER and symbiosomes in bid1 suggests that host protein secretion may be impaired.

217 GFP fusions of NCR001 [no italics for this and the other proteins listed]

217 symbiosome is misspelled

217 of the DNF1

223 Confocal microscopy revealed that while

223 tagged is misspelled
247 the ER
248 the ER
248 spreading throughout the cytoplasm. Using AutoCUTs-SEM
249 structure is closely associated with differentiating symbiosomes
253 web, and in
253 associations, are
255 that function in
258 delete 'more'
258 The ER
259 organelle and the plasma
260 their direct
265 is staved the correct term?
267 and 268 the ER
275 have a broader
278 and their microbial partner in the symbiosomes.
435 shorter in length
437 maximum symbiosome length
449 of the target mutation

504 for the secretion pathway in nodule cells.

Response to the comments of reviewers:

Reviewer #1 (Remarks to the Author):

This manuscript explores the role of BID1, a putative peptide peptidase, suggested to be involved in cleaving signal peptides in the ER. A genome rearrangement mutant, *bid1*, was isolated and nicely used to illustrate the consequences of impaired BID1 function in ER dynamics, protein secretion and nodule function.

The strength of this research is that it represents a novel approach to understand ER-symbiosome association and the role of BID1 in protein secretion required for symbiosome maturation. A major weakness is that the central claim of BID1 being a peptide peptidase is only supported by indirect evidence like sequence homology and AF2 modelling. Biochemical results supporting the peptidase activity of BID1 would make the story substantially stronger.

Response: We thank Reviewer #1 for the objective and fair assessment of our manuscript. Indeed, it is important to demonstrate the biochemical activities of BID1, especially for identifying BID1 substrates beyond SP (signal peptide) fragments.

We conclude that BID1 is a SPP (signal peptide peptidase) based on several evidences. First, *SPP* is a housekeeping gene in eukaryotic cells, it is conserved among all kingdoms, making them highly recognizable and their functions general predictable. In the case of *Medicago* BID1, it has unusually high sequence similarity with SPPs from other species, in Supplemental Figure 3 of our manuscript, we show that the sequence similarity between BID1 and human SPP is 43%, which is high for a plant protein. The amino acids critical reported to be critical for SPP protease activities are not substituted in BID1 protein. Second, we show that BID1 specifically localizes to ER membrane, making it an ER membrane-specific aspartyl protease with high sequence similarity with human SPP. By AlphaFold2 protein structural prediction, we find that the predicted structures of BID1 is identical to human SPP. BID1 is a protein with 8 trans-membrane domains, so it will be difficult to use prokaryotic protein purification experimental system to demonstrate its biochemical functions. To us an *in vitro* protein expression and a lipid membrane reconstituted experimental systems are required for the assays of BID1 biochemical activities, especially for identifying its direct substrates, but this work probably is more suitable for another independent study.

Overall, we agree with Reviewer #1 that although it is highly convincing that BID1 is a SPP, it is important to demonstrate the biochemical activities of BID1 directly in future studies.

Expression levels of ExoY, ExoB, CtrA and BacA were higher in *bid1* mutants (Fig 1) and this is suggested to indicate a free living state of the bacteroids. Not sure what is the evidence for this higher expression is related to the free-living state. This statement needs to be substantiated or supported by references if this has been shown before.

Response: We thank Reviewer #1 for the valuable suggestion. *exoY* and *exoB* are key genes regulating the initiation of exopolysaccharide synthesis, several transcriptomic studies showed that *exoY* is expressed more highly in free-living bacteria and the nodule tip than in mature bacteroids (Barnett et al., Proc Natl Acad Sci U S A, 2004, doi: 101(47):16636-41; Roux et al., Plant J., 2014, doi: 10.1111/tpj.12442). Thus, we reasoned the down-regulation of *exoY* and *exoB* indicated a state more like free-living conditions regarding exopolysaccharide production. *ctrA* is a master regulator of bacterial cell cycle, it has been reported that the expression of *ctrA* is significantly down-regulated in bacteroids once differentiation begins (Roux et al., Plant J., 2014, doi: 10.1111/tpj.12442), and the protein level of *ctrA* is much lower in mature bacteroids (Pini et al., Mol Microbiol. 2013, doi: 10.1111/mmi.12347). The expression of *bacA* gene is also activated during bacteroid differentiation (Lang et al., Front Plant Sci. 2018, doi: 10.3389/fpls.2018.00076). Bacteroids with *bacA* mutated fail to differentiate and senesce soon after their release into nodule cells (Glazebrook et al., Genes Dev., 1993, doi: 10.1101/gad.7.8.1485).

To make the point more clearer, we changed this sentence to “The expression levels of *exoY*, *exoB*, *ctrA* and *bacA* were much higher compared with WT (Fig. 1i, 1j, Supplementary Fig. 1f, 1g), indicating altered states of exopolysaccharides synthesis and impaired differentiation of bacteroids in *bid1* cells”. Here we cited Barnett et al., Proc Natl Acad Sci U S A, 2004, doi: 101(47):16636-41, Roux et al., Plant J., 2014, doi: 10.1111/tpj.12442 and Lang et al., Front Plant Sci. 2018, doi: 10.3389/fpls.2018.00076 here. The sentence is in Line 96 of the revised manuscript.

Reviewer #2 (Remarks to the Author):

This manuscript describes a signal peptide peptidase required for nodule function. The phenotype is very strong, with blocked symbiosomes differentiation and N-fixation. The cell biology is of high quality, and the most important conclusions are strongly supported (ie. that in *bid1* nodules the ER is dissociated from the symbiosomes). The nodulation phenotype presumably results to a defect in ER function resulting either from the accumulation of signal peptides in ER. The gene is single copy in *Arabidopsis* its loss is lethal. Hence, *BID1* represents a unique opportunity to study signal peptide peptidases in plants, an issue that is discussed by the authors.

I have only a few minor points and a list of suggested edits which mostly concern grammar.

The authors conclude since only legumes with indeterminate nodules have BID1 that genes function in secretory pathway evolved nodule-specific duplicates frequently. However, they show that BID1 is limited to a specific legume clade, strongly suggesting that BID1 evolved once, in the common ancestor of the IRLC clade. The conclusion is therefore incorrect, at least based on the data presented. Also, mimosas, a non-IRLC- legume, form indeterminate nodules. Is BID1 present in mimosas?

Response: We thank Reviewer #2 for the critical assessment of this point. In our original manuscript we wanted to highlight that in legumes a group of genes were generated specifically to function in nodule-specific secretory pathway, especially in IRLC-clade species. We believe the duplication of these genes are important, thus mention the phenomenon multiple times in the manuscript.

Medicago genome has a housekeeping BID1 homolog, BID1L. In our phylogenic assay, BID1 is specific to IRLC-clade legumes, while BID1L could be found in all legume species. Some species, e.g., soybean, have more than 1 SPP orthologs in their genomes, however, they are clearly orthologs of BID1L. Following the Reviewer's advise, we analyzed the orthologs of BID1 and BID1L in the genome of *Mimosa pudica*, two copies of BID1L orthologs could be found, but not BID1. This result shows that BID1 is probably specific to IRLC-clade species, but not legumes forming indeterminate nodules.

We also built a phylogenetic tree of DNF1 and DNF1L (see below), which has not been done before. SPC22 (name of the Signal Peptidase Complex component *DNF1* encodes) orthologs could be found in non-legume dicot and monocot species. In legumes, DNF1 orthologs are also specific to IRLC-clade species, while other legumes only have orthologs of DNF1L. Thus, the evolvement history of DNF1 and BID1 are quite similar, they are probably both evolved once in IRLC-clade legumes specifically.

In the revised manuscript, we state that evolution of BID1 happened once in IRLC-clade species specifically. The descriptions of gene duplication events of genes function in secretory pathway in legumes are also changed, please check Line 158 to 162, Line 714 to 716 of the revised manuscript.

minor comments:

The thickness of optical section used for confocal images should be noted, since when considering associations between the ER and bacteroids, thick sections could be misleading.

Response: We thank Reviewer #2 for the kind advise. We agree that during confocal microscopy, thick sections would increase the background noise signals and be misleading. In our assays, the nodules were hand-sectioned in half, the sectioned products were cut in half again for the observation. The thickness varied but were generally less than 2 mm. To minimize the possible background noises, we used the smallest pinhole sizes when possible, to reduce the length of axial resolution, during the confocal microscopy assay. And as shown in the manuscript, we also determined the proximal associations between ER and symbiosomes through TEM assays and 3D remodelling. We have added a sentence to the material and method section of our manuscript, at Line 347 of the revised manuscript, reading “For confocal microscopy, nodules were hand-sectioned in half, the products were sectioned again for the observations”.

In Fig 5, is the signal peptide usually ER membrane associated? If so this should be shown in the diagram.

Response: We thank the reviewer for the critical evaluation of our model. Signal peptide is required to insert the nascent, newly synthesized proteins from ribosomes into the ER membrane. Upon releasing from nascent protein sequences by the Signal Peptidase Complex, on the ER membrane the signal peptide fragments will be degraded by Signal Peptide Peptidase, to ensure proper function of the ER. Thus, signal peptide is strongly associated with ER membrane. Following the Reviewer's advise, in our model we moved the signal peptide fragments to the membrane of the ER. Please check Figure 5 of the revised manuscript.

The authors say that the GFP tag needed be inserted at N-terminus of BID1. You must provide details; did you try C-term and it failed?

Response: We thank the reviewer for the careful assay of our results. BID1 has a "KKXX" motif at the C termini of its amino acid sequence, this motif could modulate the localization onto ER membrane. And it has been demonstrated by many studies that "KKXX" motif need to be resident at the C termini, otherwise the proteins would fail to localize to ER membrane.

In our manuscript, in Figure 2b (the complementation assay) we show that when the GFP tag was inserted into the C termini of BID1, the *pBID1::gBID1-GFP* construct could not complement *bid1*, while *pBID1::GFP-gBID1*, the construct with GFP tag at the N termini could. GFP-BID1 fusion protein localized to the ER membrane specifically. We did not check the subcellular localization of BID1-GFP, as the mis-localized protein was not functional, and was likely to be subjected to degradation in nodule cells.

In extended data Fig 1, indicate what ND stands for

Response: We thank the reviewer for the valuable suggestion. "ND" means "not detected". We mentioned "ND" in Figure 1h and Supplementary figure 1, revisions have been made in both places. Please check Line 574 and 669 of the revised manuscript.

In Extended data Fig9c, were these genes expressed from pBID1? Please make this more clear.

Response: We thank the reviewer for the kind advise. In this manuscript we used NCR001, NCR166 and CAML, 3 proteins that are known to be secreted to symbiosome as markers, to evaluate the defects of host protein secretion in *bid1* mutant. *pBID1* was chosen to express these markers, as in *bid1* nodule cells the *BID1* promoter could drive the expression levels of these markers to levels similar to WT, or even higher. We revised the figure legend of Supplementary Fig. 9c to "c, When

expressed under *BID1* promoter, the expression levels of *NCR001-GFP*, *NCR166-GFP* and *CAML-GFP* were even slightly higher in *bid1* compared with WT. 14 dpi *pBID1::NCR001-GFP*, *pBID1::NCR166-GFP* and *pBID1::CAML-GFP* expressing WT and *bid1* nodules inoculated with *ABS7 hemA::LacZ* were used for qRT-PCR assay.” The sentence is now in Line 762 to 766 of the revised manuscript.

Extended In Extended Data Figure 4. The species name is missing for lcu_2RBY

Response: We thank the reviewer for pointing this error out. Please check the revised Supplementary Figure 4.

Abstract line 27: implications of what?

Response: We thank the reviewer for the valuable suggestion. We wanted to state that the function and structural adaption of ER in nodule cells were neglected, and research into key genes controlling functions and adaptations of ER in nodule cells can improve our understanding of sub-cellular interactions of ER with symbiosomes and other organelles. However, due to the word limit of the Abstract (150 words), we have very limited space here. Thus, we changed the sentence to “However key components controlling the adaption of ER in nodule cells remain elusive.” We hope the Reviewer find the revision acceptable. Please check Line 24 of the revised manuscript.

29: to the ER membrane

Response: Corrected. Please check Line 27 of the revised manuscript.

30: the ER

Response: Corrected. Please check Line 28 of the revised manuscript.

31: but not in bid1

Response: Corrected. Please check Line 29 of the revised manuscript.

58: of the NF symbiosis

Response: Corrected. Please check Line 56 of the revised manuscript.

62 membrane-to-membrane structures for interaction with other

Response: Corrected. Please check Line 59 of the revised manuscript.

62 and the plasma membrane

Response: Corrected. Please check Line 60 of the revised manuscript.

63 the ER

Response: Corrected. Please check Line 61 of the revised manuscript.

64 delete aforementioned or use 'the aforementioned'

Response: Thanks. We changed the sentence to “with the aforementioned host protein secretion as the most prominent example”. Please check Line 62 of the revised manuscript.

67 at the ER

Response: Corrected. Please check Line 65 of the revised manuscript.

69 studies suggest that

Response: Corrected. Please check Line 67 of the revised manuscript.

70 adjustments during rhizobial

Response: Corrected. Please check Line 69 of the revised manuscript.

73 to assert the delicate control of

Response: Corrected. Please check Line 69 of the revised manuscript.

74 to SPs, upon SP excision, it remains unclear how nodule cells cope with SP fragments on the ER membrane.

Response: Corrected. Please check Line 70-72 of the revised manuscript.

87 membrane is necessary (delete comma)

Response: Corrected. Please check Line 74 of the revised manuscript.

87 Confocal microscopy and TEM assays confirmed that in *bid1* cells, rhizobia within symbiosomes failed to differentiate

Response: Corrected. Please check Line 85 of the revised manuscript.

154 In *Medicago*, *BID1* has one homolog with a highly similar sequence.

Response: Corrected. Please check Line 153 of the revised manuscript.

154 Provide the gene model for BIDL

Response: We thank the reviewer for the kind advise. The gene model of *BID1L* is in Supplementary Fig. 3b of the revised manuscript. Also we added a sentence to describe the results in Line 154 of the revised manuscript, reading “The gene structure of BID1L is also highly similar to BID1 (Supplementary Fig. 3b)”.

160 delete specifically

Response: Thanks for the advise. Here we changed the sentence into “BID1 paralogs were specific to IRLC-clade species only”, to describe the evolvement of BID1 more correctly. Please check Line 159 of the revised manuscript.

161 This is similar to nodule-specific DNF1

Response: Corrected. Please check Line 160 of the revised manuscript.

184 in bid1 nodule cells

Response: Corrected. Please check Line 183 of the revised manuscript.

187 In bid1, the ER

Response: Corrected. Please check Line 186 of the revised manuscript.

187 Suggested rewording: In bid1, the ER was arranged irregularly throughout the nodule (Extended Data Fig. 6).

Response: We thank the reviewer for the kind advise. The sentence was changed to “ In *bid1*, the ER was arranged irregularly throughout the nodules (Supplementary Fig. 6), indicating failed ER structural reconfiguration.” Please check Line 186-187 of the revised manuscript.

190 an mCherry

Response: Corrected. Please check Line 189 of the revised manuscript.

190 structure instead of arrangement

Response: Corrected. Please check Line 189 of the revised manuscript.

192 only in the infection

Response: Corrected. Please check Line 191 of the revised manuscript.

194 and were in close proximity to GFP-labeled symbiosomes,

Response: Corrected. Please check Line 193 of the revised manuscript.

195 and the ER failed to form regularly arrayed webs around the symbiosomes.

Response: Corrected. Please check Line 194 of the revised manuscript.

197 form an extensive

Response: Corrected. Please check Line 196 of the revised manuscript.

203 structures were disordered and failed to associate closely with symbiosomes

Response: Corrected. Please check Line 201 of the revised manuscript.

207 with an average distance between the ER and symbiosomes of several dozen nanometers

Response: Corrected. Please check Line 206 of the revised manuscript.

208 refs 13,14 belong is the corresponding section in the discussion

Response: We thank Reviewer #2 for the kind advise. The two references have been cited in the corresponding part of the discussion. Please check Line 260 of the revised manuscript.

209 failed to surround [encircle is 2D, surround is 3D]

Response: We thank the Reviewer for the kind suggestion. Please check Line 208 of the revised manuscript.

212-13 Suggested wording: Our results suggest that the ER of N-fixing nodule cells is closely associated with symbiosomes, and that this requires BID1.

Response: We rewrote the sentence following the advise of the reviewer. Please check Line 210-212 of the revised manuscript.

216 The dissociation of ER and symbiosomes in bid1 suggests that host protein secretion may be impaired.

Response: We rewrote this sentence following the advise of Reviewer #2. Please check Line 215 of the revised manuscript.

217 GFP fusions of NCR001 [no italics for this and the other proteins listed]

Response: Corrected. Please check Line 217-218 of the revised manuscript.

217 symbiosome is misspelled

Response: Corrected. Please check Line 216 of the revised manuscript.

217 of the DNF1

Response: Corrected. Please check Line 217 of the revised manuscript.

223 Confocal microscopy revealed that while

Response: Corrected. Please check Line 223 of the revised manuscript.

223 tagged is misspelled

Response: Corrected. Please check Line 223 of the revised manuscript.

247 the ER

Response: Corrected. Please check Line 247 of the revised manuscript.

248 the ER

Response: Corrected. Please check Line 248 of the revised manuscript.

248 spreading throughout the cytoplasm. Using AutoCUTs-SEM

Response: Corrected. Please check Line 249 of the revised manuscript.

249 structure is closely associated with differentiating symbiosomes

Response: Corrected. Please check Line 249 of the revised manuscript.

253 web, and in

Response: Corrected. Please check Line 252 of the revised manuscript.

253 associations, are

Response: Corrected. Please check Line 253 of the revised manuscript.

255 that function in

Response: Corrected. Please check Line 255 of the revised manuscript.

258 delete 'more'

Response: Done. Please check Line 258 of the revised manuscript.

258 The ER

Response: Corrected. Please check Line 258 of the revised manuscript.

259 organelle and the plasma

Response: Corrected. Please check Line 259 of the revised manuscript.

260 their direct

Response: Corrected. Please check Line 260 of the revised manuscript.

265 is staved the correct term?

Response: We thank the reviewer for the suggestion. We changed the sentence into “as is the case of phosphatidylethanolamine transportation between ER and mitochondria in mammalian cells with the nutrients depleted”. Please check Line 266 of the revised manuscript.

267 and 268 the ER

Response: Corrected. Please check Line 267-268 of the revised manuscript.

275 have a broader

Response: Corrected. Please check Line 275 of the revised manuscript.

278 and their microbial partner in the symbiosomes.

Response: Corrected. Please check Line 277 of the revised manuscript.

435 shorter in length

Response: Corrected. Please check Line 567 of the revised manuscript.

437 maximum symbiosome length

Response: Corrected. Please check Line 569 of the revised manuscript.

449 of the target mutation

Response: Corrected. Please check Line 628 of the revised manuscript.

504 for the secretion pathway in nodule cells.

Response: Corrected. Please check Line 649 of the revised manuscript.

Besides the revisions mentioned here, in the Figures and Supplementary Information data points were added to the plots when applicable. We revised the Title, Abstract, and Legend of Figure 3, mainly to meet the format standards of Nature Communications. All the revisions were colored in red. We thank both Reviewers for their professional and generous work to improve our manuscript.

REVIEWERS' COMMENTS

Reviewer #1 (Remarks to the Author):

The rebuttal letters discussion of the evidence for BID1 being a signal peptide peptidase and the difficulties foreseen for the biochemical characterization as well as substrate identification is much appreciated. Adding a couple of sentences to the discussion section touching on these aspects would add perspective to the work presented and help readers.

Reviewer #2 (Remarks to the Author):

The revisions look complete and the manuscript has been improved.

I just have some small edits:

Abstract: "However, the key components controlling..."

L59 "The ER forms direct..."

L70 control (instead of controls)

L71 related to SPs; upon SP excision,

L159 delete 'only'

L160 frequently means several times, so unless you think there are other components with nodule specific duplicates I suggest the following rewording:

"..suggesting that IRLC-clade legumes evolved these nodule-specific duplicates to promote terminal symbiosome differentiation."

Response to Reviewers' comments

Reviewer #1 (Remarks to the Author):

The rebuttal letters discussion of the evidence for BID1 being a signal peptide peptidase and the difficulties foreseen for the biochemical characterization as well as substrate identification is much appreciated. Adding a couple of sentences to the discussion section touching on these aspects would add perspective to the work presented and help readers.

Response: We thank Reviewer #1 for the kind suggestions. We have revised a sentence to mention this aspect, reading "Further investigations into BID1, particularly through the identification of its direct substrates by biochemical assays, will significantly contribute to fully revealing the molecular communications between nodule host cells and their microbial partner in the symbiosomes". Please check Line 277 of the revised manuscript.

Reviewer #2 (Remarks to the Author):

The revisions look complete and the manuscript has been improved.

I just have some small edits:

Abstract: "However, the key components controlling..."

Response: Corrected. Please check Line 24 of the revised manuscript.

L59 "The ER forms direct..."

Response: Corrected. Please check Line 59 of the revised manuscript.

L70 control (instead of controls)

Response: Corrected. Please check Line 70 of the revised manuscript.

L71 related to SPs; upon SP excision,

Response: Corrected. Please check Line 71 of the revised manuscript.

L159 delete 'only'

Response: Corrected. Please check Line 159 of the revised manuscript.

L160 frequently means several times, so unless you think there are other components with nodule specific duplicates I suggest the following rewording:

"..suggesting that IRLC-clade legumes evolved these nodule-specific duplicates to promote terminal symbiosome differentiation."

Response: Corrected. Please check Line 160-162 of the revised manuscript.